# Ta Doping Effect on Structural and Optical Properties of InTe Thin Films

**DOI:** 10.3390/nano10091887

**Published:** 2020-09-21

**Authors:** Chunmin Liu, Yafei Yuan, Xintong Zhang, Jing Su, Xiaoxiao Song, Hang Ling, Yuanjie Liao, Hao Zhang, Yuxiang Zheng, Jing Li

**Affiliations:** 1Department of Optical Science and Engineering, Shanghai Ultra-Precision Optical Manufacturing Engineering Center, Fudan University, Shanghai 200433, China; 18110720032@fudan.edu.cn (C.L.); 15110720031@fudan.edu.cn (Y.Y.); 17110720006@fudan.edu.cn (X.Z.); 16110720001@fudan.edu.cn (J.S.); 19110720013@fudan.edu.cn (X.S.); 18210720094@fudan.edu.cn (H.L.); 19210720009@fudan.edu.cn (Y.L.); zhangh@fudan.edu.cn (H.Z.); yxzheng@fudan.edu.cn (Y.Z.); 2Department of Electronic Engineering, Center for Intelligent Medical Electronics, Fudan University, Shanghai 200433, China

**Keywords:** Ta-doped InTe thin film, mixed-valence compound, linear optical absorption, Z-scan, two-photon absorption

## Abstract

The objective of this work was to study the influence of Ta doping on the structural, transmittance properties, linear absorption parameter, and nonlinear absorption properties of InTe thin films. The as-deposited samples with different Ta doping concentrations were prepared by a magnetron co-sputtering technique and then annealed in nitrogen atmosphere. Structural investigations by X-ray diffraction revealed the tetragonal structure of InTe samples and that the crystallinity decreases with increasing Ta doping concentration. Further structural analysis by Raman spectra also showed good agreement with X-ray diffraction results. The Ta doping concentration and sample thickness determined by energy-dispersive X-ray spectroscopy and scanning electron microscopy increased as Ta dopant increased. In addition, X-ray photoelectron spectroscopic was carried out to analyze the chemical states of the elements. UV–VIS–NIR transmittance spectra were applied to study the transmittance properties and calculate the linear absorption coefficient. Due to Burstein–Moss effect, the absorption edge moved to shorter wavelengths. Meanwhile, the values of band gap were found to increase from 1.71 ± 0.02 eV to 1.85 ± 0.01 eV with the increase of Ta doping concentration. By performing an open aperture Z-scan technique, we found that all Ta-doped InTe samples exhibited two-photon absorption behaviors. The nonlinear optical absorption parameters, such as modulation depth, two-photon absorption coefficient, and two-photon absorption cross-section, decrease with increasing Ta concentration, whereas the damage threshold increases from 176 ± 0.5 GW/cm^2^ to 242 ± 0.5 GW/cm^2^. These novel properties show the potential for applications in traditional optoelectronic devices and optical limiters.

## 1. Introduction

Chalcogenide thin films have drawn considerable interest in recent years, owing to their novel properties and broad range of applications in high-performance optical and electric fields, such as phase change random-access memory (PRAM), resistance random-access memory (ReRAM), nonlinear optical absorption (NOA) devices, thin-film solar cells, and so on [1,2,3,4]. Among them, group IIIA metal chalcogenide (GIIIAMC) is an important member. On the basis of stoichiometric composition, GIIIAMC can be simply divided into two categories: One is AB type, such as InSe, InTe, GaSe, and GaTe, and the other is A_2_B_3_ type, for example, In_2_Se_3_, In_4_Se_3_, and In_2_Te_3_ [5]. As mentioned, indium telluride (InTe), better written as In^1+^In^3+^Te_2_^2−^, is a mixed-valence compound belonging to the AB type with a chain TlSe-type structure (space group I4/mcm) [6]. The ionic and covalent substructures coexist in this structure, that is, the trivalent In^3+^ ions are tetrahedrally coordinated by four Te^2−^ ions, forming covalent In–Te bonds. On the other hand, the monovalent In^1+^ ions are surrounded by eight Te^2−^ ions in a tetragonal antiprismatic arrangement [7]. The two different crystallographic positions of distinct In^3+^ and In^1+^ prevent the free transfer of electrons from In^1+^ to In^3+^ [8].

For applications, Jana et al. reported that the room temperature lattice thermal conductivity of InTe is as low as 0.76 W/(m·K), which is much smaller than those of other Te-based compounds, such as 2.9 W/(m·K) of GeTe [9]. This characteristic is beneficial to thermoelectric material development since it can be applied to the thermoelectric solid state cooling technologies and waste heat power generation [10]. For optical absorption performance, Peng et al. calculated that the upper limit of the energy conversion efficiency of sunlight of the InTe reached as large as 6.4%, which was superior to those of the transition metal dichalcogenides (TMDs) with a predicted energy conversion efficiency of less than 0.4% [11,12]. In the field of PRAM, the contaminants induced errors and low phase-change temperature (< 453 K) limited the usage of some compounds such as Ge–Sb–Te and Ge_2_Sb_2_Te_5_ [13]. Sugiyama et al. reported the phase-change temperature of InTe thin films over 523 K, which overcomes such a disadvantage [14]. Especially, Yuan et al. found that InTe, a promising optical limiting material, has superior third-order nonlinear characteristics [15]. However, the change of InTe damage threshold is not clear. Although InTe has been widely used in both switching devices and solar cells, there is still some opportunities for better performance, such as forming heterojunctions with other materials, adjusting the proportion of elements, and doping foreign elements [16,17,18]. Among them, doping is a common method. For instance, several Ti-doped chalcogenides have been well studied in theory and experiments, but little has been reported about other inert metal elements [15,19]. In the present work, tantalum, an inert metal element, is chosen as the dopant because it has a higher solid solubility and can induce high electrical conductivity while preserving high transmittance [20,21]. For example, Ta has been used as an outstanding candidate for improving stability and thermal properties of chalcogenide thin films [22]. In addition, though many studies relating to the growth, structural characteristics, pressure dependent behaviors, and thermoelectric properties of InTe films have been reported [6,7,9,18], there are few studies on the NOA properties of intrinsic or doped InTe.

Therefore, combining these two considerations, the Ta element is chosen as the dopant to extensively study the effects on the structural, transmittance, linear optical absorption, and NOA properties of InTe thin films. To gain insights into the adjustability of InTe films with the Ta dopant, various properties of Ta-doped InTe films, such as the crystal structure, optical band gap, nonlinear absorption parameters, etc., are discussed in-depth, as follows. The results are important for using the InTe material in many applications.

## 2. Experimental Details

A LAB600sp-type magnetron co-sputtering system (Leybold Optics GmbH, Dresden, Germany), was adopted to prepare film samples. Through magnetron co-sputtering, the Ta-doped InTe thin films was deposited on fused quartz and Si (100) substrates with the same conditions. The sputtering power of the InTe target was set to constant value of 200 W, and that of the Ta target was changed from 0 to 25 W. The background and working pressures in the chamber were controlled at 7 × 10^−6^ mbar and 2.8 × 10^−3^ mbar, respectively. The purity of both targets is 99.99%. All the substrates were kept at room temperature during the whole sputtering process. After deposition, a set of samples was annealed at 450 °C, for 30 min, in a flowing nitrogen atmosphere.

The crystallization characteristics of the samples were determined by X-ray diffraction (XRD) with the Cu-Kα (1.54056 Å) radiation (Bruker D8 ADVANCE) (Karlsruhe, Germany), in the 2*θ* angles range of 10–60°, with a step of 0.02°. The scanning electron microscopy (SEM) (Hillsboro, Oregon, USA) mapping scan and energy dispersive X-ray spectroscopy (EDS) (EDAX Inc., New Jersey, USA) elemental analysis were performed via an FEI Siron 200 ultra-high resolution scanning electron microscope. The maximum magnification is × 10^6^, and the resolution is 1 nm @ 1 kV. Raman spectra of the samples were tested by using the Nanofinder 30(Tokyo Instruments, Tokyo, Japan). A double-beam UV–VIS–NIR spectrophotometer (Shimadzu UV-3600) (Shimadzu Co. Ltd., Kyoto, Japan), country) was employed to determine the transmission spectroscopic characteristics. Chemical components and bonding analyses were carried out by using X-ray photoelectron spectroscopy (XPS; Kratos Axis Ultra Dld) (Shimadzu Co. Ltd., Kyoto, Japan).

In order to measure the NOA properties of Ta-doped InTe thin films, a single beam open-aperture (OA)-mode Z-scan method was built for it. The details of the setup were discussed in our previous work [3]. All the samples were excited by a pulsed laser at 800 nm excitation wavelength with 100 fs pulse duration and 1 kHz repetition rate from the Ti-sapphire regenerative amplifier system (Spectra Physics, Spitfire Ace) (California, USA). In this system, the Rayleigh length ($Z_{0}=\frac{{\pi \omega }^{2}}{\lambda }$) and the beam radius, ω, at the focal point were 4 mm and 32 μm, respectively. Each sample thickness was much smaller than the diffraction length of focused beam. Therefore, the samples could be regarded as “thin” films [23]. In the process of measurement, each thin-film sample was moved along the Z axis (propagation direction), to be driven by a stepping motor. Through calibration tests, our system met the requirements of Z-scan theory [3].

## 3. Results and Discussion

### 3.1. XRD Analysis

Four annealed Ta-doped InTe thin film samples are named S1–S4, with increasing deposition powers of Ta target, respectively. Figure 1a shows the XRD patterns of the S1–S4 samples. The sharp diffraction peaks suggest high crystallinity and the polycrystalline nature of the films. All the samples show the characteristic peaks of tetragonal phase of InTe (JCPDS No: 30–0636) with lattice parameters a = b = 8.454 Å and c = 7.152 Å, indicating the formation of better crystalline InTe films. This structure is consistent with some researches [6,7]. The absence of Ta diffraction peaks indicates that the Ta dopant is well mixed in the lattice of InTe; similar results have been reported by previous work [24].

As presented in Figure 1b, S1–S4 exhibit a preferred (211) orientation, and the intensity of this diffraction peak decreases with the increase of the Ta dopant concentration, while the full width at half maximum (FWHM) of the (211) peak increases accordingly. In addition, Table 1 gives the variations of the FWHM and crystallite sizes. The mean crystallite size (*D*) is estimated from the Scherrer Equation [25]:(1)$$D=\frac{0.9\lambda }{B\cos\theta }$$
where *λ* is the wavelength of X-ray radiation (1.54056 Å), *B* represents the full width at half maximum of the peak, and *θ* refers to the angle of Bragg diffraction. The calculated values of crystallite sizes decrease with increasing Ta dopant concentration. It indicates that the increase of Ta dopant concentration will reduce the crystallinity of the film samples. It should be attributed to stress, as well as the formation of defects or disorders due to segregation of dopants on the grain boundaries [24].

### 3.2. Raman Spectral Analysis

As an additional characterization, the Ta-doped InTe thin films were studied by Raman spectroscopy, and the results are shown in Figure 2. There are five vibrational bands located at ~102, ~125, ~142, ~194, and ~267 cm^−1^. Notable peaks for these samples and phonon modes are listed in Table 2. Our Raman spectrum of the intrinsic InTe film matches well with that in other works [26,27]. As shown in Figure 2, the two weak peaks ~194 and ~267 cm^−1^ gradually disappear as the concentration of the dopant increases. This observation could be attributed to the destruction of the long Te chain by the doping of Ta. Similar experimental phenomena have been reported in the literature [15]. The peak fitting results of the three main peaks are shown in Figure 3. The phonon modes at about 125 and 142 cm^−1^ are assigned to A_1g_ and E_g_ symmetry, respectively. The phonon assignment matches well with the previously reported Raman studies of InTe by Nizametdinova [28]. The A_1g_ mode is due to the motion of Te atoms alone, while for E_g_ mode, Te and In^3+^ atoms vibrate in opposite directions. Based on the work of Rajaji et al. [7], in these two modes, the In^1+^ atoms do not exhibit any movement. Besides, the phonon mode at about 102 cm^−1^ is assigned to lower E_g_ mode, which agrees with the Raman spectra recorded by Pine et al. [26]. Comparing S1–S4, the phonon mode at about 142 cm^−1^ moves to about 138 cm^−1^ after Ta doping, and the intensity decreases with the increase of Ta dopant concentration. This should be due to the destruction of translational symmetry as a result of generation of defects or substitution. 

### 3.3. SEM, EDS, and XPS Analysis

The film thickness estimated from SEM images of S1–S4 are 306 ± 1, 327 ± 1, 338 ± 1, and 356 ± 1 nm, respectively. The SEM images of S2 and S3 are shown in Figure 4. In the SEM images, the ambient, main Ta-doped InTe layer, and Si substrate are clearly distinguished. In addition, it should be noted that the protrusions in the SEM image for S3 are the marginal remnants left behind when cutting the sample. The composition of the Ta-doped InTe thin films from EDS are listed in Table 3. Obviously, both the thickness and the weight percentage of Ta dopant increase as the sputtering power of the Ta target increases. 

The X-ray photoelectron spectroscopic measurement was performed to analyze the chemical states of the samples. The 3D high-resolution spectrum of S3 in Figure 5a shows two strong peaks at ~444.4 and ~452.0 eV, which can be assigned to In 3d_5/2_ and In 3d_3/2_ with a spin-orbit splitting of 7.6 eV. Similarly, as seen in Figure 5b, two strong peaks corresponding to Te 3d_5/2_ and Te 3d_3/2_ states are observed at ~572.5 and ~582.9 eV, respectively. The XPS results of In and Te are consistent with the InTe crystal [31]. Four prominent Ta 4f states are observed in Figure 5c. The former peaks of Ta 4f_7/2_ at ~26.3 eV and Ta 4f_5/2_ at ~28.3 eV are attributed to Ta^5+^ state, which should be introduced by the resonance bond. Resonance happens when more than one valence-bond structure can be written for a molecule or ion. Then, the true structure is a blend of all the different possible structures [32]. In this work, the combination of Ta and Te elements forms a complex structure; thus, Ta^5+^ state appears.

The latter peaks at ~22.5 eV (Ta 4f_7/2_) and ~24.4 eV (Ta 4f_5/2_) correspond to a Ta^4+^ state, indicating the formation of TaTe_2_. The same experimental results were reported in a previous study [33]. These results suggest that the Ta element forms both Ta^5+^ and Ta^4+^ chemical states when Ta is doped into the InTe film.

### 3.4. Linear Optical Analysis

The optical transmittance spectra of the Ta-doped InTe thin films in the spectrum range of 300–3000 nm are displayed in Figure 6a. The transmittance of the thin-film samples in the visible region is much smaller than that in the near-infrared region. Specifically, the transmittance is almost 84% in the transparence region, and it starts to decrease sharply for wavelengths shorter than 1090 nm. The value of transmittance increases after Ta doping; however, the transmittance decreases slightly if the concentration of the Ta dopant continues to increase. The results indicate that Ta doping can improve the transmittance, and the observed decrease of transmittance could be due to the increased absorption of free carriers and the enhanced photon scattering of impurities as the Ta doping level increases [21]. In addition, the absorption edge of the samples shifts toward shorter wavelength (blue-shift) with the increase of Ta doping concentration, implying the increase of band gaps. A related analysis is given in the subsequent calculation section.

The linear absorption coefficient, *α*, is a key parameter for the characterization of optical band gap and NOA mechanism. It can be calculated from the following Equation [34]:(2)α=−ln(T)/d
where *T* is the transmittance, *d* is the thickness of the Ta-doped InTe thin film. The calculated linear absorption spectra are plotted in Figure 6b. In accordance with the transmittance spectra, the values of linear absorption coefficient, α, decrease slightly after Ta doping. The absorption region covers the entire visible region and extends up to the near-infrared region, which can be attributed to the presence of different density of localized defect states in the gap.

Based on Tauc’s method [35], the optical band gap (*E_g_* ) of the Ta-doped InTe film samples can be calculated by using Equation (3):(3)$$\left(\alpha h\nu \right)=A{\left(h\nu -E_{g}\right)}^{n}$$
where,α is the calculated linear absorption coefficient from transmittance spectra, *hv* refers to the incident photon energy, *A* is a constant which is related to the effective mass and *n* is a constant that determines the type of optical transitions. For direct transitions in InTe with a narrow gap, n is equal to 1/2 [36]. The optical band gap (*E_g_*) can be derived from the interception of the linear part of (*αhv*)^2^ vs. hv for (*αhv*)^2^ = 0. As plotted in Figure 7, the sizes of band gap for S1–S4 are 1.71 ± 0.02, 1.78 ± 0.01, 1.80 ± 0.01 and 1.85 ± 0.01 eV, respectively. It is clear that the optical band gap increases with the increase of the Ta doping. Similar experimental results had also been reported in our previous work, which may be attributed to the Burstein–Moss effect [15,37]. That is, the increase of carrier concentration of the doped films and filling of the electronic states in the conduction band can increase the band gap.

### 3.5. Nonlinear Optical Analysis

The ultrafast NOA properties of Ta-doped InTe thin films were investigated by a single-beam OA Z-scan method with femtosecond laser pulses at the wavelength of 800 nm [3]. In the experiment, the samples were sequentially moved along the Z-axis, and the values of intensity-dependent transmittance were recorded. All the samples were measured at a relative low incident power density of 99 GW/cm^2^, to avoid nonlinear scattering. According to the relationship between transmittance and incident laser intensity, the NOA behaviors in semiconductors can be divided into two types. In a nutshell, saturable absorption (SA) behavior occurs if transmittance increases with increasing incident laser intensity (symmetric peak). The other type shows a symmetric valley, namely, the transmittance decreases as the incident laser intensity increases. This optical limiting effect includes two-photon absorption (TPA), multiphoton absorption, and reverse saturable absorption (RSA) [38].

Figure 8 depicts the NOA behaviors of S1–S4, where the dots are experimental data and the curves represent the best fitting results. It is worth noting that the reversible NOA behaviors are caused by the intense laser light, rather than the structural changes of samples or the NOA properties of fused quartz substrates. It can be seen that the normalized transmittance gradually decreases as the sample position approaches the laser focal, whereas the normalized transmittance gradually increases as it moves away from the beam focus, i.e., all the samples exhibit symmetric valleys. This observation is attributed to TPA, since the band gap energy of Ta-doped InTe samples (1.71–1.85 eV) is larger than the energy of one photon (~1.55 eV), but smaller than the energy of two photons (~3.10 eV) [39]. For different samples, the transmittance at the beam focus shows a slight increase at a higher Ta doping concentration. For simplification, the band structure of the Ta-doped InTe thin films is employed to simulate this absorption process, as shown in Figure 9a. An electron in the valence band cannot be directly excited from the valence band to the conduction band by absorbing a photon, because the energy of one single photon is less than the band gap. However, an electron in the valence band can be excited to the conduction band by absorbing two photons. 

The typical OA Z-scan theory by Sheik-Bahae was used to determine the TPA coefficient, *β*. It can be deduced by the following equation [23]:(4)$$T_{OA}=\sum_{m=0}^{\infty }\frac{{\left[-\beta I_{0}L_{eff}/\left(1+z^{2}/z_{0}^{2}\right)\right]}^{m}}{{\left(m+1\right)}^{3/2}}$$
where *T_OA_* is the normalized transmittance of the OA Z-scan curve, *I_0_* represents the incident laser intensity at the focal point, *z* indicates the position of the thin films on the axis, *z_0_* is the Rayleigh length, and *L_eff_* is the effective thickness of samples, which can be calculated by the following [3]:(5)$$L_{eff}=\frac{1-e^{-\alpha_{0}L}}{\alpha_{0}}$$
with *α_0_* and *L* as the linear absorption coefficient at the wavelength of 800 nm and the physical thickness of the sample, respectively. The resultant fitting parameters are listed in Table 4, and the TPA coefficient decreases with the increase of Ta doping concentration. At 800 nm excitation wavelength, the nonlinear absorption coefficients of In_2_Te_3_, Sb_2_Se_3_, WS_2_, and MoS_2_ are 805.6, 843.67, −397, and −136.13 cm/GW, respectively [3,40,41,42]. The values of the TPA coefficient are comparable to those of similar chalcogenides. For application purposes, the damage thresholds of Ta-doped InTe films were investigated and illustrated in Figure 9b. The damage thresholds increase from 176 ± 0.5 GW/cm^2^ to 242 ± 0.5 GW/cm^2^ with increasing Ta doping concentration. This is beneficial to the application in laser-damage-resistance devices. It is worth noting that the doping of Ta increases the damage threshold, while maintaining the high TPA coefficient of InTe material. The TPA behaviors of Ta-doped InTe thin films with large nonlinear absorption coefficient and damage threshold at room temperature are highly advantageous for applications in optical power limiters. Furthermore, the TPA cross-section, *σ_2_*, can be calculated by the following [43]:(6)$$\sigma_{2}=\frac{h\nu \beta }{N_{A}d_{0}\times 10^{-3}}$$
where *hv* refers to the incident photon energy, *N_A_* is the Avogadro’s number, and *d_0_* is the molar concentration of the absorbing molecules (in units of M). The TPA cross-section, *σ_2_*, has a unit of cm^4^/(photon/s) or cm^4^ s. It can also be represented by another informal unit, GM, which is defined by the following:(7)$$1GM=10^{-50}{\mathrm{cm}}^{4}s$$

The calculated data are also listed in Table 4. Clearly, the value of *σ_2_* decreases with increasing Ta doping concentration, which is consistent with the experimental observation. According to He et al., most of the measured values of *σ_2_* are in the range from 10^−51^ to 10^−46^ cm^4^ s [43]. The values of *σ_2_* are within this reasonable range.

## 4. Summary

In the present work, Ta element was chosen as the dopant to improve the performance of the chalcogenide film. The microstructure, transmittance characteristics, linear optical absorption parameters, and nonlinear optical absorption properties of the InTe films with different Ta doping concentrations were investigated and discussed in detail. These samples were prepared on both fused quartz and Si substrates by a magnetron co-sputtering method, and then they were annealed at 450 °C for 30 min. X-ray diffraction measurement indicates a crystalline state for all samples, and the crystallinity decreases with increasing Ta doping concentration. Raman spectra analysis also shows good agreement with X-ray diffraction results. The thickness of the samples increases at increasing Ta doping concentration. Chemical analysis reveals that the Ta dopant exists in the InTe films as Ta^5+^ and Ta^4+^ states. The transmittance measured via UV–VIS–NIR improves after Ta doping, but it will decrease by further increasing the Ta concentration. Due to the Burstein–Moss effect, the calculated values of optical band gaps increase from 1.71 ± 0.02 eV to 1.85 ± 0.01 eV as the Ta doping concentration increases. The NOA responses of the Ta-doped InTe thin films were investigated by using the OA Z-scan technique with the excitation of 800 nm. All samples show two-photon absorption effect. The NOA parameters, such as modulation depth, TPA coefficient, and TPA cross section decrease with increasing Ta doping concentration. In addition, the damage threshold increases at higher Ta dopant concentrations. The performance of InTe films can be adjusted according to Ta doping concentration. Overall, InTe film with low doping concentration could be employed in innovative optoelectronic devices and optical limiters in view of its high transmittance and nonlinear absorption coefficient.

## Figures and Tables

**Figure 1 nanomaterials-10-01887-f001:**
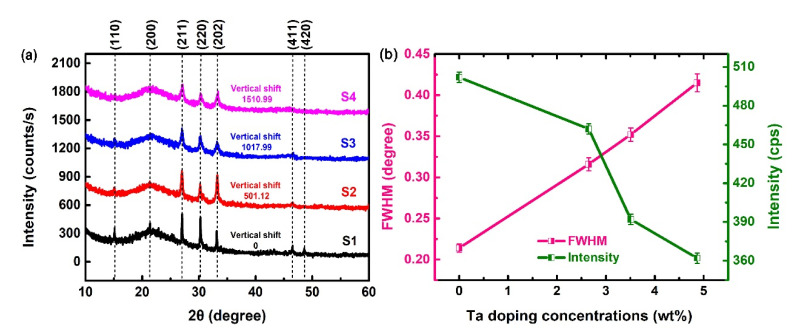
(**a**) XRD patterns of Ta-doped InTe thin films S1–S4; (**b**) variations of FWHM and intensity of the (211) peak.

**Figure 2 nanomaterials-10-01887-f002:**
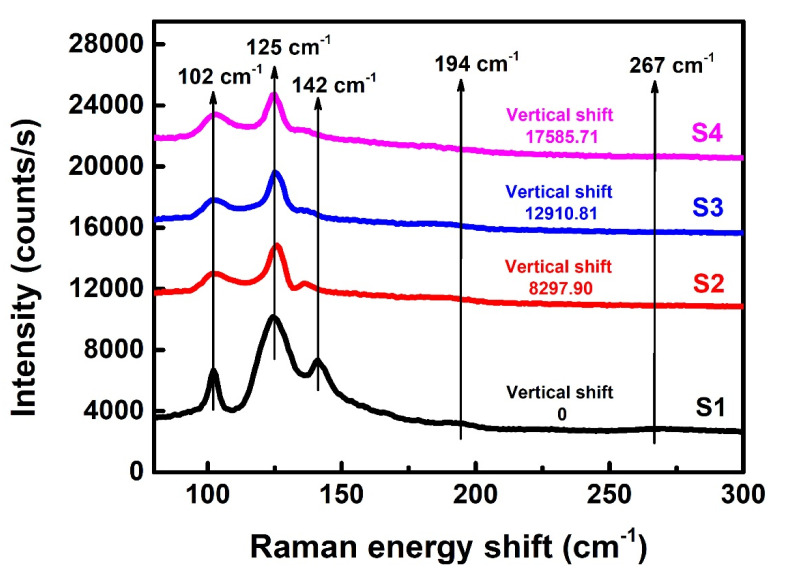
Raman spectra of Ta-doped InTe thin films S1–S4.

**Figure 3 nanomaterials-10-01887-f003:**
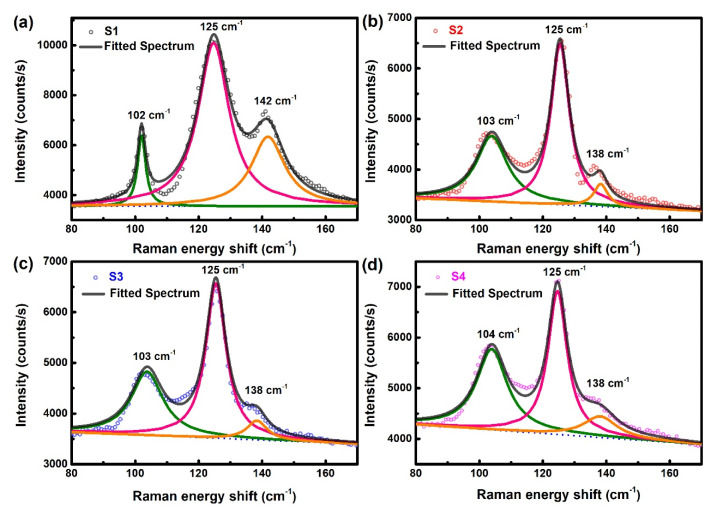
The peak fitting Raman spectra of Ta-doped InTe thin films: (**a**) S1, (**b**) S2, (**c**) S3, (**d**) S4

**Figure 4 nanomaterials-10-01887-f004:**
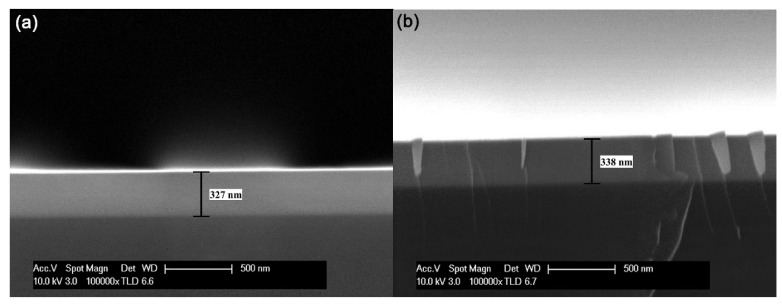
The cross-sectional SEM images of (**a**) S2 and (**b**) S3.

**Figure 5 nanomaterials-10-01887-f005:**
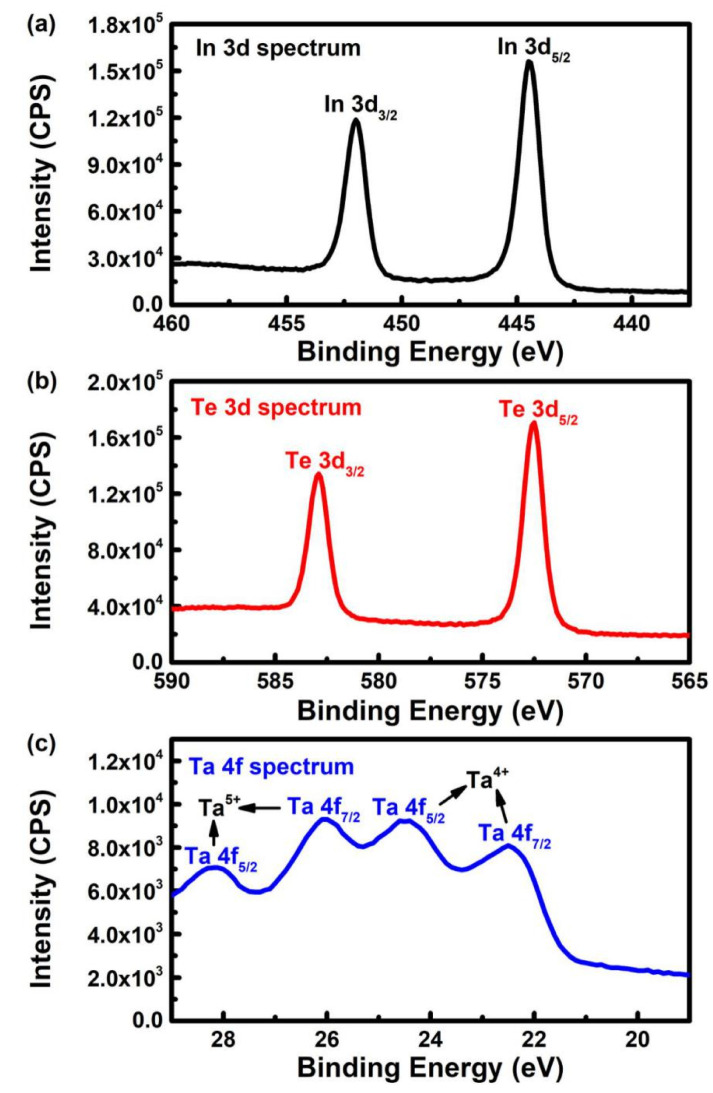
XPS spectra of S3: (**a**) In 3d, (**b**) Te 3d, and (**c**) Ta 4f regions.

**Figure 6 nanomaterials-10-01887-f006:**
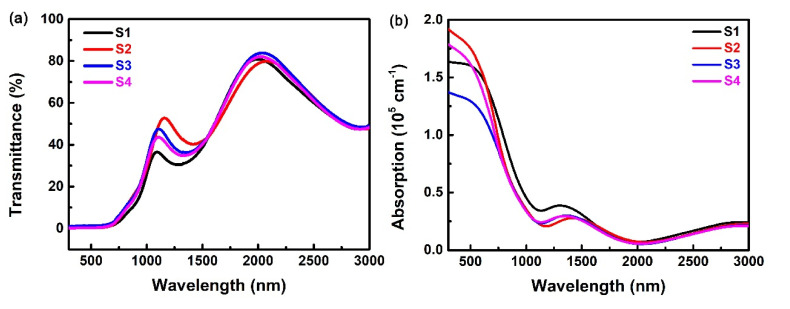
The (**a**) transmittance and (**b**) absorption spectra of Ta-doped InTe thin films S1–S4.

**Figure 7 nanomaterials-10-01887-f007:**
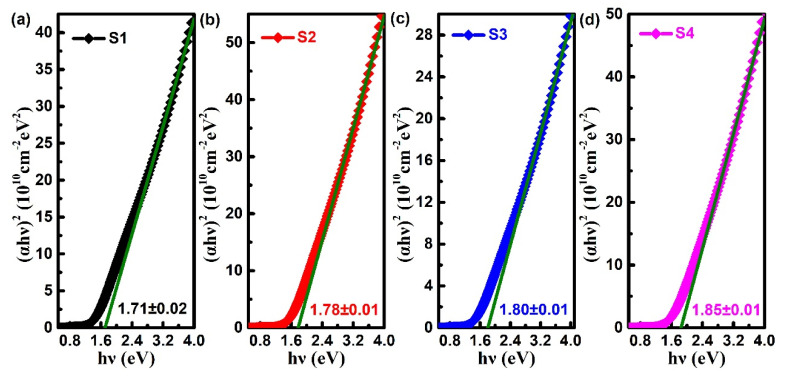
The optical band gaps of Ta-doped InTe thin films: (**a**) S1, (**b**) S2, (**c**) S3, (**d**) S4.

**Figure 8 nanomaterials-10-01887-f008:**
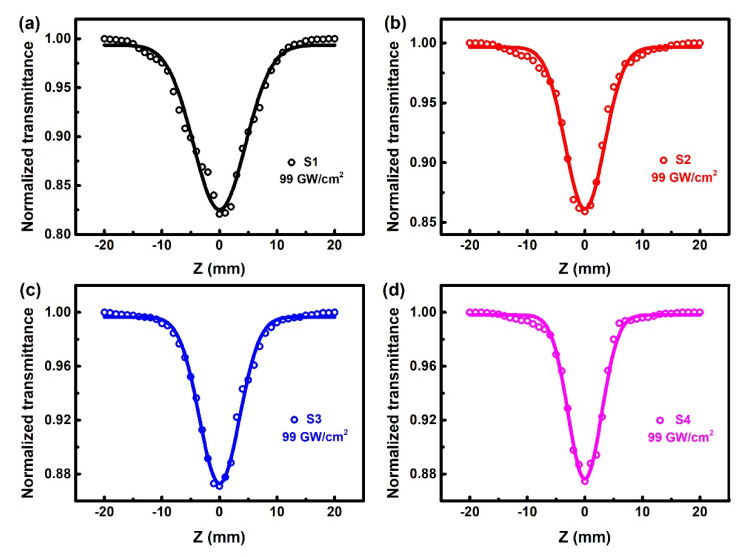
The open-aperture (OA) Z-scan curves of Ta-doped InTe thin films (**a**) S1, (**b**) S2, (**c**) S3, (**d**) S4 at the excitation of 800 nm wavelength.

**Figure 9 nanomaterials-10-01887-f009:**
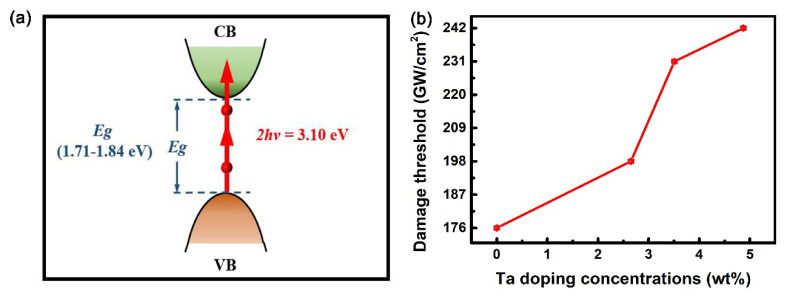
(**a**) The two-photon absorption (TPA) process and (**b**) damage threshold of Ta-doped InTe thin films.

**Table 1 nanomaterials-10-01887-t001:** Preparation parameters and the variations of the full width at half maximum (FWHM) and crystallite sizes.

Sample	InTe Power (W)	Ta Power (W)	FWHM (degree)	Crystallite Size (nm)
S1	200	0	0.2142 ± 0.0052	38.2 ± 0.8
S2	200	15	0.3167 ± 0.0078	25.8 ± 0.6
S3	200	20	0.3520 ± 0.0079	23.2 ± 0.5
S4	200	25	0.4144 ± 0.0110	19.7 ± 0.5

**Table 2 nanomaterials-10-01887-t002:** Raman peaks of Ta-doped InTe thin films and related phonon modes.

Reference	Characteristic Wavenumbers (cm^−1^)									
This Work	--	--	102 ± 1	125 ± 1	142 ± 1	--	--	194 ± 1	--	267 ± 1
Te [26]	--	--	105	123	143	--	--	--	--	--
Te [27]	--	--	--	120	140	--	176	--	--	260
InTe [28]	46	86	--	126	139	--	--	--	--	--
In_2_Te_3_ [29]	--	--	103	125	142	157	--	194	225	--
CuInTe_2_ [30]	--	--	--	123	143	--	171	--	220	267

**Table 3 nanomaterials-10-01887-t003:** The composition and thickness results of Ta-doped InTe thin films.

Sample	Ta Power (W)	In (wt.%)	Te (wt.%)	Ta (wt.%)	Thickness (nm)
S1	0	51.07	48.93	0	306 ± 1
S2	15	50.19	47.16	2.65	327 ± 1
S3	20	48.12	48.37	3.51	338 ± 1
S4	25	49.31	45.81	4.87	356 ± 1

**Table 4 nanomaterials-10-01887-t004:** Nonlinear optical absorption (NOA) parameters of the Ta-doped InTe thin films measured by OA Z-scan.

Sample	*L_eff_* (nm)	*β* (cm/GW)	*σ_2_* (GM)
S1	97.4 ± 0.05	526.40 ± 0.27	89.42 ± 0.05
S2	117.2 ± 0.08	343.55 ± 0.22	58.35 ± 0.04
S3	125.5 ± 0.09	293.68 ± 0.20	49.89 ± 0.03
S4	127.0 ± 0.07	281.88 ± 0.17	47.89 ± 0.03

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
