# Peer review of "Ta Doping Effect on Structural and Optical Properties of InTe Thin Films"

_nanomaterials, 2020, doi:10.3390/nano10091887_

Round 1
Reviewer 1 Report
In this paper, the influence of Ta doping into InTe thin film on optical properties and structure is investigated. Four samples (S1-S4) with different Ta concentration are prepared by magnetron co-sputtering technique and annealing in nitrogen ambient, then these property are compared. think the importance of InTe and the reason why you select Ta as a dopant is briefly explained in this manuscript. However, I do not think the superiority of this material compared with other possible materials is enough mentioned.
1) Please explain the superiority of this material by comparing with other materials in more detail.
2) How are the device performances are improved by utilizing the obtained optical property in this material ?
Reviewer 2 Report
Dear Authors and Editor,
in the manuscript "Ta doping modification effect on structural and optical properties of InTe thin films" by C. Liu et al. the influence of the Ta doping to the structural and optical properties of the InTe films is reported. This manuscript contains a systematic study at various doping concentration and provides valuable physical quantities for these doped thin films. Despite that the information provided in the manuscript can be very meaningful for the material research community, the following issues must be satisfactorily addressed before the manuscript could be published:
(1) The error bars for the data in Fig.1(b), Tab.1-4, Fig.9(b), as well as for the fitted values of the band gap in Fig.7 must be provided.
(2) On page 6, the meaning of the last sentence in section 3.2 is not clear. How does x-ray diffraction show that the dopants inhibit the growth of grains? And why this should lead to a reduction of the Raman signals?
(3) In Fig.2, a correct physical unit for the intensity is required.
(4) On page 8, in the last sentence, the meaning of the "resonance bond" is not clear. Please kindly explain it in one sentence and provide appropriate references.
(5) Are the non-linear optical parameters in Tab.4 reasonable? A comparison with existing materials or modelling can be greatly helpful.
(6) Several additional minor corrections can be found in the attachment.

Reviewer 3 Report
Authors analyse the effects of doping InTe thin films with Ta on the physical structure of the films and on their linear and nonlinear absorption properties.
Whereas their study seems correct I believe that significant changes must be perform before publication.
i. Authors stands that they analyse linear optical properties and nonlinear absorption. However, linear optical properties of an optical medium are characterized by the refractive index and the absorption coefficient. There are not any mentioned to the refractive index. Authors can argue that they also study transmission properties, but they mainly depend on absorption.
ii. When presenting results, authors often refers to other studies and this is not clear what is original in the present study. They must clarify it.
iii. In the introduction section, they are mentioned two operating pressures in films preparation. Do these different pressures refer to fused quartz and silica substrates? If it is the case, why are the pressure values so different?
iv. SEM measurements accounts for sample thickness from 306 to 356 nm. It will be nice to give some data about measurement resolution.
v. In fig. 4 the SEM image of sample S2 seems quite homogeneous while the image for sample S3 clearly shows some structures. Some comments are advisable.
vi. In section 3.4 authors affirm that the samples transmittance shows obvious interference patterns. In my opinion, interference is not clear. Transmission modulations can be due to the illumination pattern and sensibility of the spectrometer.
vii. Furthermore interference, if present, is due to multiple reflection in the film surfaces. But this is not considered in eq. (2) when calculating the absorption coefficient.
vii. After eq. (2) it is written: “ in contrast to the transmission spectra, the values of the linear absorption coefficient decrease…” But transmission and absorption are related magnitudes, so “In accordance to”, or similar, must substitute “in contrast to”.
viii. In fig. 6a, the behaviour of S1 at lower wavelengths is quite strange. I imagine that it is due to some errors in the very low transmission of the samples at these wavelengths.
ix. Simple calculation show that the measured nonlinear absorption is an order of magnitude lower than linear absorption. This could be noted.
Round 2
Reviewer 2 Report
Dear Editor and the Authors,
the manuscript has been extensively improved and could be published if the following questions are properly addressed:
(1) Methods for estimating the error bars in Tables 1, 2, 3, and 4 need to be described in detail. Especially for the Raman peak positions in Table 2 as well as the sample thickness in Table 3, it seems to be unreasonable that all the uncertainties have the same values and they are also too small. As a comparison for the width of diffraction peaks in Table 1, it will be greatly helpful if the Authors can provide the angular resolution of the diffraction experiments.
(2) As a reply to my third question, the Authors wrote: "For a clearer display, the Raman curves of the four samples are divided longitudinally in Fig.2. Therefore, the specific value of the ordinate is meaningless and we didn't list the value of the ordinate." It will be informative if the Authors can indicate values on the y-axis of Fig.2 and mention the amount of the vertical shifts for each curve in Fig.2. Similarly, the scale of the vertical axis in Fig.1(a) needs to be provided with a proper physical unit.
(3) In the reply to the first question of the reviewer 3, the Authors report new data for the index of refraction. However, the imaginary part of the index is missing. It will be greatly helpful if the Authors can provide the complete complex index of refraction.
Reviewer 3 Report
In my opinion, the manuscript can be accepted in the present form
